# Metabonomics and Transcriptomic Analysis of Free Fatty Acid Synthesis in Seedless and Tenera Oil Palm

**DOI:** 10.3390/ijms25031686

**Published:** 2024-01-30

**Authors:** Lu Wei, Cheng Yang, Jerome Jeyakumar John Martin, Rui Li, Lixia Zhou, Shuanghong Cheng, Hongxing Cao, Xiaoyu Liu

**Affiliations:** 1Coconut Research Institute, Chinese Academy of Tropical Agricultural Sciences, Wenchang 571339, China; wl79912021@163.com (L.W.); 15730095249@163.com (C.Y.); jeromejeyakumarj@gmail.com (J.J.J.M.); lirui@catas.cn (R.L.); glzz_2009@163.com (L.Z.); 2National Key Laboratory for Tropical Crop Breeding, Haikou 571101, China; 3School of Horticulture, Hainan University, Haikou 570228, China; 4College of Tropical Crops, Yunnan Agricultural University, Pu’er 665000, China; 1999033@ynau.edu.cn

**Keywords:** oil palm, metabolomics, transcriptomics, lipid synthesis, free fatty acid

## Abstract

Oil palm, a tropical woody oil crop, is widely used in food, cosmetics, and pharmaceuticals due to its high production efficiency and economic value. Palm oil is rich in free fatty acids, polyphenols, vitamin E, and other nutrients, which are beneficial for human health when consumed appropriately. Therefore, investigating the dynamic changes in free fatty acid content at different stages of development and hypothesizing the influence of regulatory genes on free fatty acid metabolism is crucial for improving palm oil quality and accelerating industry growth. LC-MS/MS is used to analyze the composition and content of free fatty acids in the flesh after 95 days (MS1 and MT1), 125 days (MS2 and MT2), and 185 days (MS3 and MT3) of Seedless (MS) and Tenera (MT) oil palm species fruit pollination. RNA-Seq was used to analyze the expression of genes regulating free fatty acid synthesis and accumulation, with differences in genes and metabolites mapped to the KEGG pathway map using the KEGG (Kyoto encyclopedia of genes and genomes) enrichment analysis method. A metabolomics study identified 17 types of saturated and 13 types of unsaturated free fatty acids during the development of MS and MT. Transcriptomic research revealed that 10,804 significantly different expression genes were acquired in the set differential gene threshold between MS and MT. The results showed that FabB was positively correlated with the contents of three main free fatty acids (stearic acid, myristate acid, and palmitic acid) and negatively correlated with the contents of free palmitic acid in the flesh of MS and MT. ACSL and FATB were positively correlated with the contents of three main free fatty acids and negatively correlated with free myristate acid. The study reveals that the expression of key enzyme genes, FabB and FabF, may improve the synthesis of free myristate in oil palm flesh, while FabF, ACSL, and FATB genes may facilitate the production of free palmitoleic acid. These genes may also promote the synthesis of free stearic acid and palmitoleic acid in oil palm flesh. However, the FabB gene may inhibit stearic acid synthesis, while ACSL and FATB genes may hinder myristate acid production. This study provides a theoretical basis for improving palm oil quality.

## 1. Introduction

Oil palm, a highly versatile crop, can be categorized into two primary resource types: African oil palm (*Elaeis guineensis* Jacq.) and American oil palm (*E. oleifera*, previously *E. melanococca*). African oil palm is the most widely cultivated and commercially significant variety, known for its high oil yield and adaptability to different climatic conditions. On the other hand, American oil palm is less commonly grown and is mainly utilized for breeding purposes to develop new hybrid varieties with improved traits [1]. Oil palm is widely cultivated in Malaysia, Indonesia, Nigeria, and other countries has brought substantial commercial value. The large-scale cultivation of oil palm has proven to be highly profitable due to its high oil yield and versatility in various industries. However, at present, some countries in Southeast Asia expand the planting area of oil palm by reducing the area of forest, which directly poses a certain threat to biodiversity, reduces the habitat of organisms, and affects the formation of the greenhouse effect. Therefore, it is very important to maintain a good balance between the planting area of oil palm and the forest cover area, which is related to whether to maximize the economic value of oil palm under the premise of protecting the environment and biodiversity [2,3,4]. The classification of oil palm is based on the thickness of the fruit shell, which can be divided into three types: Dura, Pisifera, and Tenera. This categorization helps in understanding the different characteristics and uses of each type. In addition to the mentioned matrilineal varieties, there are also paternal varieties of certain crops. For instance, in the case of certain crops like Deli dura and Bamenda, they are primarily matrilineal varieties. On the other hand, Ekona and Nigeria are examples of paternal varieties. These different varieties have their own distinct characteristics, including traits such as yield, resistance to diseases, and adaptability to specific environmental conditions.

China’s decision to import oil palm in the 1920s was a significant move that involved importing a substantial amount of germplasm from countries like Indonesia. This decision had far-reaching implications for China’s agricultural landscape and economy. By importing oil palm, China was able to diversify its crop production and tap into the economic benefits associated with this versatile plant. Currently, the oil palm has been introduced and cultivated in various provinces in China, including Hainan, Yunnan, Guangdong, Guangxi, and Fujian [5]. The seedless oil palm variety mentioned in this study is a result of hybridization between O×G Amazon species and contains higher unsaturated fatty acids. The Tenera oil palm is a variety produced by Deli×Ghana hybridization and has a higher oil content in its fruits. Therefore, both varieties are good experimental materials and can be used for correlation analysis of their fatty acids.

Oil palm is one of the most productive tropical crops globally, with the highest oil production [6]. The oil is extracted from the fruit, known as Crude Palm Oil (CPO) [7]. CPO, or Crude Palm CPO, is not only edible, but also plays an important role in various industries. It is utilized in the production of cosmetics, detergents, chemicals, and pharmaceuticals [8,9]. CPO is mainly composed of saturated fatty acids with a significant portion of palmitic acid and oleic acid and unsaturated fatty acids, with oleic acid and linoleic acid being the most prominent ones. In addition to fatty acids, it also contains vitamins, fatty alcohols, carotenoids, and other nutrients [10]. The content of fatty acids in flesh and the content ratio of different components directly affect the yield and quality of CPO. The current focus of oil palm biotechnology research is to enhance the levels of unsaturated fatty acids such as oleic acid and linoleic acid in flesh [11], so as to improve the oil yield of oil palm fruit and the quality of CPO. In addition, the content of free fatty acids in palm oil serves as a crucial quality indicator for various aspects of its storage, marketing, and production time, as well as the price of palm oil products [12], and the high content of free fatty acids leads to poor fat quality [13]. Therefore, the quality of palm oil is closely related to the content of free fatty acids.

The synthesis of oil palm fatty acids is the most important phase in oil formation and serves as the basis for yield formation. Improving palm oil yield can be achieved by mining genes and transcription factors related to oil palm fatty acid synthesis and analyzing their regulatory effects [14,15,16]. At present, the genes related to fatty acid synthesis of oil palm have been studied. For instance, Cytochrome P450 is a superfamily of genes involved in various biochemical reactions [17,18], and plays an important role in plant fatty acid metabolism. According to Liang et al. [19], the concentration of fatty acids in the peel of oil palm increases consistently throughout its development, reaching its highest point at maturity. This similarity in expression pattern P450 genes in the same tissue, indicating that cytochrome P450 expression impacts the metabolism of oil palm fatty acids (oxidation, epoxidation, hydrocarbylation, etc.), thereby influences the composition and content of oil palm fatty acids. During the rapid accumulation of fatty acids in African oil palms, a recent study found a significant increase in the expression levels of EgSAD1 and EgSAD2. Meanwhile, the hybrid oil palms exhibited a notably higher content of unsaturated fatty acids than the African oil palms. The differential expression of OeSADs in hybrid oil palms may contribute to the increased unsaturated fatty acids in these palms. This suggests that the high expression of SAD genes could potentially influence the fatty acid composition of hybrid oil palms [20]. The overexpression of EgFatB1, EgFatB2, and EgFatB3 in oil palm was successfully transferred to wild Arabidopsis Thaliana. This transfer revealed that the overexpression of these genes resulted in an increased palmitic acid content in transgenic seeds. EgFatB3 is an important enzyme that positively regulates medium-chain fatty acid synthesis. This enzyme is responsible for the production of new fatty acids, specifically lauric acid and myristate acid. Interestingly, out of the two, myristate acid tends to accumulate the most. It is speculated that EgFatB1, EgFatB2, and EgFatB3 in oil palm regulate the synthesis of palmitic acid, and EgFatB3 promotes the synthesis of medium-chain fatty acids [21].

The study aims to analyze the changes in free fatty acid content and key enzyme gene expression in Seedless (O×G Amazon) and Tenera (Deli×Ghana) oil palm fruits after pollination. The changes in free fatty acid content of two fruit varieties were examined at different time points after postharvest. The study reported on the levels of free fatty acids at 0 h, 24 h, and 36 h., Additionally, candidate genes that may play a role in regulating these changes in free fatty acid content were identified [22]. By combining metabolomics and transcriptomics analysis of changes in free fatty acids during the development stage of oil palm fruits is crucial in understanding the metabolic processes that occur during this period., By identifying the key genes involved in the production of free fatty acids, valuable insights into the factors that influence oil accumulation in oil palm fruits can be gained. This knowledge can then be utilized to improve oil palm breeding programs and enhance oil production in these two varieties. Therefore, understanding the dynamic changes of free fatty acids during the fruit development stage is an important step towards maximizing the potential of oil palm cultivation and meeting the growing demand for palm oil.

## 2. Results

### 2.1. Analysis of Dynamic Changes in Metabolites of Free Fatty Acids in the Flesh of Seedless and Tenera Oil Palm Species at Different Developmental Stages

After pollination, the mesocarp of oil palm fruits at different stages, namely 95d (MS1 and MT1), 125d (MS2 and MT2), and 185d (MS3 and MT3), were selected for metabolomic analysis. Three biological replicates were performed at each stage to ensure the reliability and accuracy of the results. Qualitative mass spectrometric analysis of sample metabolites was performed based on information obtained from a local lipid database, after which mass spectrometric peaks observed in different samples for each substance were corrected to ensure accurate quantification. The quantitative analysis of a randomly selected substance in different samples was integrated with the correction results (Appendix A). By performing overlapping display analysis on the total ion flow chromatogram (TIC) of mass spectrometry detection and analysis of the same quality control sample (Appendix A), the total ion flow chromatogram (TIC) of mass spectrometry detection and analysis of QC samples showed significant overlap. The CV value distribution map of the QC samples reveals that more than 75% of substances have CV values less than 0.3 (Appendix A), exhibiting that the metabolomics data are relatively stable and reliable. Principal component analysis (PCA) was used to segregate the six sets of data, namely MS1, MS2, MS3, MT1, MT2, and MT3, including quality control samples (Appendix A). The first principle component (PC1) and second principal component (PC2) were reported to account for 57.35% and 13.1% of the total variables, respectively.

The research on the free fatty acid metabolites in the flesh of Seedless and Tenera varieties at different development stages revealed that the main constituents were saturated free fatty acids. In fact, the study identified a total of 17 different types of saturated free fatty acids, such as palmitic acid (C16:0), stearic acid (C18:0), decanoic acid (C10:0), and unsaturated free fatty acids, such as oleic acid (C18:1), linoleic acid (C18:2), linolenic acid (C18:3), and 13 other components (Table 1). As shown in Table 1, the total free fatty acid content of both the Seedless and Tenera varieties experienced a gradual increase as the flesh grew and developed. However, it is worth noting that there was a significant spike in the early to middle stages of growth. The oleic acid content in the flesh undergoes significant changes as it matures, with the highest level of free fatty acid content in ripe flesh. The data show a significant increase in the concentration of the substance from MS1 to MS3, with a rise from 277.30 nmol/g to 32,992.80 nmol/g. Similarly, in the MT group, the concentration increased from 4549.25 nmol/g in MT1 to 30,396.86 nmol/g in MT3. These findings suggest a substantial increase in the amount of the substance over time in both the MS and MT groups. According to the results, the linoleic acid content in Seedless oil palm pulp was found to be the least in MS1, the highest in MS3, and the next-highest in terms of free fatty acid content. However, Tenera oil palm pulp had the least amount of palmitic acid in MT1, the most in MT3, and the second highest amount in term of its free fatty acid content. As the Seedless oil palm fruit ripens from MS1 to MS3 and MT1 to MT2, the levels of palmitic acid in the flesh increase significantly. In fact, the palmitic acid content reaches its peak during this stage, making it one of the highest levels of free fatty acids found in the fruit. During the ripening stage, the Teneral oil palm’s flesh experiences a significant increase in the amount of linoleic acid.

The research findings indicate that Seedless oil palm has a distinct advantage over MS2 and MS3 in terms of its lower total free fatty acid content. Additionally, Seedless oil palm showcases a significantly higher percentage of saturated free fatty acid, reaching an impressive 75.69%. These results suggest that Seedless oil palm may be a more desirable option due to its reduced levels of free fatty acid and its higher proportion of saturated free fatty acid when compared to MS2 and MS3. The unsaturated to saturated free fatty acid (UFFA/SFFA) ratio of 0.33 was found to be lower than MS2 and MS3. MT1 exhibited a lower total free fatty acid level compared to MT2 and MT3. Furthermore, the ratio of unsaturated to saturated free fatty acid content (UFFA/SFFA) in MT1 was 2.57, which was lower than that of MT2 but higher than MT3. The data presented in Figure 2 show a gradual accumulation of palm oil in both the Seedless and Tenera palm species. The concentration of free fatty acids in the Seedless oil palm has experienced a significant increase. It was 30.24 times the total content of all kinds of free fatty acids in the MS1 stage. Moreover, the percentage of unsaturated free fatty acids in the total content reached 77.96%. Additionally, the ratio of unsaturated to saturated free fatty acid (UFFA/SFFA) was recorded at 3.54, further highlighting the dominance of unsaturated fatty acids in the MS1 stage. The total content of free fatty acids in Tenera oil palm significantly increased to 45746.14 nmol/g. This was found to be 4.91 times higher than the total content of all kinds of free fatty acids observed in the MT1 stage. The percentage of unsaturated free fatty acids reached its peak at 72.70%. Additionally the ratio of unsaturated and saturated free fatty acids (UFFA/SFFA) reached the maximum at 2.67. The total content of free fatty acids in Seedless oil palm showed a significant increase, reaching 58,231.41 nmol/g, which was 32.84 times higher than the total content of all kinds of free fatty acids in MS1 and 1.09 times higher than in MS2. The ratio of unsaturated to saturated free fatty acids (UFFA/SFFA) reached its maximum (4.48). The total content of free fatty acids in Tenera oil palm increased to 60,174.75 nmol/g, which was 6.46 times higher than the total content of various free fatty acids in MT1 and 1.32 times higher than in MT2. The percentage of saturated free fatty acids reached its peak at 31.23%, surpassing the levels found in MT1 and MT2. The unsaturated free fatty acid content of MT3 is 2.21 times higher than the saturated free fatty acid content. The results indicated that unsaturated free fatty acids were the main sources of oil synthesis in the flesh of the soon-to-mature Seedless and Tenera oil palms.

The contents of decanoic acid, linolenic acid, lauric acid, triaconic acid, and tridecanoic acid in MT1-MT3 showed a continuous decrease and reached the highest value at MT1, while the contents of tetracosanoic acid in MT1-MT3 initially showed a decrease followed by an increase. Analyzing the dynamic changes in the metabolites of free fatty acids in three different developmental periods, namely, slow accumulation (MS1, MT1), rapid accumulation (MS2, MT2), and stable accumulation (MS3, MT3), provides valuable insights into the metabolic processes occurring at each stage. By studying the fluctuations of these metabolites, researchers can gain a better understanding into how free fatty acids are synthesized and utilized during different developmental periods.

### 2.2. Analysis of Differential Metabolites of Free Fatty Acids in the Flesh of Oil Palm from Seedless Species and Tenera Species

The quantity of different metabolites in pairwise groups was studied by comparing the early, middle, and late stages of oil palm development. According to the variable importance in projection (VIP) ≥ 1, Fold Change ≥ 2 or Fold Change ≤ 0.5. A total of 19 metabolites of free fatty acids with significant differences were obtained. According to the comparison of differential metabolites in Seedless and Tenera oil palm species at different stages of pulp development (Figure 3A), there were 11 up-regulated metabolites and 1 down-regulated metabolite in the early stage (MS1 vs. MT1), 1 up-regulated metabolite and 6 down-regulated metabolites in the middle stage (MS2 vs. MT2), and 4 up-regulated metabolites and 5 down-regulated metabolites in the late stage (MS3 vs. MT3). The results show that there are notable differences in the levels of metabolites between MS1 vs. MT1 as well as MS3 vs. MT3. On the other hand, the comparison between MS2 and MT2 reveals a relatively low number of significantly different metabolites. Through Venn diagram analysis, it was observed that MS2 vs. MT2 and MS3 vs. MT3 exhibited the highest frequency of distinct metabolites, as depicted in Figure 3B. The results showed that the proportion of up-regulated metabolites was significantly higher than that of down-regulated metabolites, indicating that the content of most of the differential metabolites increased with the development days of flesh. In fact, it reached its maximum value in the late stage of flesh development.

### 2.3. Analysis of Differential Genes in the Flesh of Oil Palm from Seedless Species and Tenera Species

We compared samples of Seedless and Tenera oil palm pulp at the same stage of growth (MS1 vs. MT1, MS2 vs. MT2, MS3 vs. MT3). The study focused on analyzing the relationship between specific genes and their functions in the synthesis and utilization of fatty acids to uncover the essential genetic factors involved. In this study, a total of 10804 significantly differentially expressed genes were identified in oil palm pulp. These genes were obtained based on the set differential gene threshold. The differential genes in Seedless and Tenera oil palm species were analyzed at different stages of pulp development (Figure 4A). The comparison of gene expression in the early stage (MS1 vs. MT1) revealed 3159 up-regulated genes and 3981 down-regulated genes. Similarly, during pulp development (MS2 vs. MT2), there were 2395 up-regulated genes and 2486 down-regulated genes. During the later stage of the experiment, it was observed that there were 2567 up-regulated genes and 2119 down-regulated genes when comparing MS3 to MT3. Therefore, MS1 vs. MT1 had the most significantly different genes, followed by MS2 vs. MT2, whereas MS3 vs. MT3 had the least number of significantly different genes. The number of differential genes in the three comparison groups gradually decreased with the increase in flesh development days, indicating that the expression of genes regulating the growth and development of oil palm flesh in Seedless and Tenera species tended to be stable. According to Venn diagram analysis (Figure 4B), there were 1401 common differential genes among the three comparison groups.

### 2.4. Enrichment Analysis of Significantly Differentially Expressed Genes

According to the KEGG analysis, the MS1 vs. MT1 group showed enrichment in 136 pathways. Interestingly, a total of 134 pathways were found to be enriched in both the MS2 vs. MT2 and MS3 vs. MT3 groups. The findings indicated that the distinct genes in MS and MT during various growth phases primarily occur in metabolic pathways, the biosynthensis of secondary metabolites, and plant-pathogen interactions. “Metabolic pathways” and “biosynthensis of secondary metabolites” are the main metabolic pathways in plant synthesis and metabolism, and are widely involved in the biosynthesis of plant metabolites.

The enrichment degree of fatty acid synthesis pathways in the three groups of comparison follows the pattern of MS3 vs. MT3 > MS1 vs. MT1 > MS2 vs. MT2, indicating that a large number of genes are expressed in the early to middle stages of MS and MT fruit development, as well as in the mature fruit to stimulate fatty acid synthesis. Therefore, exploring the expression patterns of genes involved in the fatty acid synthesis pathway is beneficial for the further exploration of oleic acid, stearic acid, palmitic acid. The synthesis and accumulation of fatty acids such as palmitoleic acid (Figure 5).

### 2.5. Metabolomics and Transcriptomic Association Analysis

KEGG is a comprehensive database that enables the systematic analysis of metabolism and gene function [23]. The lipid metabolome and the transcriptome data were examined together using the metabolic pathway information provided by KEGG. The study revealed that 30 types of free fatty acids, including palmitoleic acid, myristate acid, arachidonic acid, tridedecanoic acid, triacsanic acid, tetracosanoic acid, docosanic acid, eicosadienoic acid, stearic acid, octadecanoic acid, palmitoic acid, and docosahexaenoic acid were enriched in nine metabolic pathways related to lipid metabolism in oil palm (Figure 6). It was shown that seven sorts of saturated free fatty acids were significantly different in the unsaturated fatty acid biosynthesis pathway between Seedless species and Tenera species, including triacontane acid, tetracosanoic acid, arachidonic acid, tridedecanoic acid, and triacontane acid, and seven corresponding differentially expressed genes. Two free fatty acids with significant differences in the fatty acid biosynthesis pathway are myristic acid and palmitoleic acid. There are two non-significant differences in saturated free fatty acids, stearic acid, and palmitoleic acid, and nineteen significantly different expression genes corresponding to these acids (Table 2).

The Nr annotation results of 19 significantly differentially expressed genes in Seedless and Tenera oil palm, revealed that the enzymes during the flesh development were 3-oxoacyl-[acyl-carrier-protein] -synthase I, FabB), 3-oxoacyl-(acyl-carrier-protein) synthaseⅡ (FabF), long-chain acyl-CoA synthetase (long-chain acyl-Coa synthetase), ACSL), and palmitoyl-[acyl-carrier-protein] -Thioesterase (FATB).

The dynamic changes in gene expression levels of four key enzymes in Seedless and Tenera oil palm species, as shown in Figure 7, revealed inconsistent patterns for the gene expression of FabB, FabF, ACSL, and FATB. The expression levels of the FabB and ACSL genes in seedless oil palm pulp showed an interesting pattern throughout the growth stages. Initially, at the beginning of flesh development, the expression of these genes was low. However, as the process progressed, the expression levels gradually increased and reached their highest point at the middle and late stage of flesh development. During the development of Tenera palm pulp, the expression level of the FabB gene consistently increased, indicating its active role in this process. On the other hand, the expression level of the ACSL gene showed a sustained decrease, suggesting a potential downregulation. Expression levels of the FabF gene and FATB gene in Seedless oil palm showed an initial decrease followed by an increase during pulp development. In contrast, the expression levels of the FabF gene and FATB gene in Tenera palm exhibited an opposite trend, with an initial increase followed by a decrease during pulp development.

The study found that the contents of four major free fatty acids in oil palm pulp were significantly correlated with the expression levels of four key enzyme genes. The contents of FabB were found to have a positive correlation with the three main free fatty acids: stearic acid, myristate acid, and palmitic acid. FabF, an enzyme involved in fatty acid biosynthesis, has been found to have a positive correlation with the levels of four key free fatty acids: stearic acid, myristate acid, palmitoleic acid, and palmitic acid. Oil palm pulp contains four main free fatty acids, and correlation analysis showed that the content of these free fatty acids was significantly related to the expression level of four major enzyme genes (Figure 7; Table 3). The study found that ACSL and FATB showed a positive correlation with the levels of three primary free fatty acids: stearic acid, palmitoleic acid, and palmitic acid. On the other hand, there was a negative correlation between ACSL and FATB and the levels of free myristate acid. These results indicate that the expression of FabB and FabF enzyme genes may enhance the synthesis of free myristate in oil palm flesh, while FabF, ACSL, and FATB enzyme genes may promote the synthesis of free palmitoleic acid. However, the expression of the FabB gene may inhibit the synthesis of stearic acid, and the expression of the ACSL and FATB gene may inhibit the synthesis of myristate acid in oil palm flesh.

### 2.6. QRT-PCR of the Transcriptomic Data

To verify the accuracy of RNA-Seq expression profile sequencing, 10 differentially expressed genes were randomly selected from related metabolic pathways and analyzed by qRT-PCR (Figure 8). The results showed that the trend of qRT-PCR was consistent with that of RNA-Seq, and the coefficient of determination (R2) was greater than 0.8.

### 2.7. Analysis of Fatty Acid Synthesis Pathways in Oil Palm Flesh at Different Development Stages

The biosynthesis of fatty acids in oil palm fruit pulp is mainly enriched in the fatty acid biological pathway (ko00061) (Figure 9; Table 2), which is composed of Acetyl CoA as raw material and ultimately forms long-chain fatty acids through the action of some enzyme genes. After the combined analysis of fatty acid transcriptome and metabolome, we were able to identify the differential metabolites and genes related to the KEGG metabolic pathway in the fatty acid biological pathway. Based on the comprehensive analysis of gene FPKM values and correlation coefficients with stearic acid, myristic acid, palmitic acid, and palmitoleic acid (Table 3), it was finally found that FATB and ACSL are significantly positively correlated with palmitoleic acid; while FabB is significantly positively correlated with stearic acid, myristic acid, and palmitic acid; FabF is significantly positively correlated with myristic acid. By combining the specific positions of these enzyme genes on the pathway, it can be concluded that FATB may promote the synthesis of palmitic acid during the MS3 phase, ACSL may promote the synthesis of palmitic acid during the MS2 and MS3 phases, and FabB and FabF may promote the formation of stearic acid, myristic acid, and palmitic acid upstream of the fatty acid synthesis pathway during the MT2 and MT3 phases. In summary, the variation in gene expression patterns between the two varieties can explain the disparities in total fatty acid content and composition. During the development process of oil palm fruit, through the combined action of the aforementioned enzymes FATB, ACSL, FabB, FabF, etc., long-chain fatty acids rich in oil palm such as stearic acid, myristic acid, palmitic acid, and palmitoleic acid are formed, thereby promoting the biosynthesis of oil palm fatty acids.

## 3. Discussions

In this study, the lipid metabolites of Seedless and Tenera oil palm pulp at three developmental stages were analyzed by liquid chromatography-tandem mass spectrometry. The results of the study reveal a significant change in the total free fatty acid content of oil palm before and after the development of the flesh, and the total free fatty acid content in the flesh gradually increases as the growth and development of the flesh progresses. In particular, there is a sharp increase in the total free fatty acid content during the early to middle stages of development. This is in line with the evolution of fatty acids in camellia fruit as the oil, oleic, linoleic, and linolenic content increase as the fruit develops [24]. The total free fatty acid content of seedless oil palm remains stable throughout its mid to late development stage. This stability is crucial as it ensures that the oil extracted from the palm retains its quality and does not degrade over time. However, the most significant changes were observed in the contents of linolenic acid, linoleic acid, palmitic acid, and eicosenoic acid in different development stages. The contents of these four free fatty acids increased by 186.95, 13,386.90, 243.92, and 395.73 nmol/g. Significant changes were observed in the flesh of Tenera seed oil palm, with an increase in the content of myristic acid, stearic acid, palmitoleic acid, and linoleic acid by 327.47, 2537.50, 92.80, and 8601.56 nmol/g, respectively. The total content of free fatty acids increased from 1733.11 nmol/g to 58,231.41 nmol/g and 9313.33 nmol/g to 60,174.75 nmol/g. However, the increase in the total content of free fatty acids in Seedless oil palm was even more substantial, with a remarkable elevation of 56,498.30 nmol/g.

FabB is a β-oxyacyl-ACP synthase that uses acyl-Coenzyme A as an intermediate to catalyze the Claisen condensation reaction between acyl-ACP and malonyl-ACP [25,26]. It is found in bacteria as part of the fatty acid synthesis II pathway and is a potential target for antibiotics [27]. Enzymes encoded by the FabA and FabB genes have been found to catalyze the introduction of double bonds in decane precursors that are elongated into the 16:1Δ9 and 18:1Δ11 unsaturated fatty acyl chains required for functional membrane phospholipids [28]. In this study the FabB enzyme gene (ID:LOC105050891) was expressed in the flesh of Seedless and Tenera oil palm species, and positively correlated with the contents of free stearic acid, myristate acid, and palmitic acid, while negatively correlated with the contents of free palmitic acid. The results indicated that the expression of the FabB gene may promote the biosynthesis of free stearic acid, myristate acid, and palmitic acid in oil palm flesh and inhibit the accumulation of free palmitic acid in oil palm flesh.

FabF is an essential enzyme for fatty acid synthesis and elongation. It plays a crucial role in the production of phospholipid membranes, lipoproteins, and lipopolysaccharides [29]. It plays a role in the FASII extension stage, where it converts malonyl ACP with growing ACP to form carbon dioxide and beta-ketoacP. Labeling it with [13C6]-glucose showed a significant decrease in the 13C flow to myristate and palmitic acid in the ΔFabB/F strain [30]. The study found that the expression of the FabF gene positively correlated with the content of free stearic acid, myristate acid, palmitic acid, and palmitic acid, suggesting that their synthesis and accumulation were promoted.

ACS can be divided into very long-chain acyl-CoA synthases (ACSVL) and long-chain acyl-Co synthases (ACSL), medium-chain acyl-Co synthases (AC-SM), and short-chain acyl-Co synthases (ACSS). ACSLs are a family of enzymes that can convert saturated and unsaturated FAs with a chain length of 8~22 into fatty acyl-CoA esters [31]. In this study, the ACSL enzyme gene showed a positive correlation with the contents of free stearic acid and palmitic acid, and a negative correlation with the contents of free myristate acid. The expression of the ACSL enzyme gene was found to have a significant impact on the fatty acid composition in the flesh of Seedless and Tenera varieties. It was speculated that the expression of this gene could promote the accumulation of free stearic acid, palmitic acid, and palmitic acid in the flesh of Seedless and Tenera varieties, and could inhibit the synthesis of free myristate acid. These findings highlight the role of the ACSL enzyme gene in influencing the fatty acid profile of these varieties and suggest its potential as a target for manipulating the fatty acid composition in agricultural crops.

Fatty acyl-ACP thioesterases in higher plants can be categorized into FATA and FATB based on amino acid sequence comparison and substrate specificity [32]. The C16:0-ACP thioesterase is a common FATB. The highest thioesterase activity was found in crude extracts of oil palm pulp [33]. Wang [34] cloned FATB from the seed kernel of LpFATB by the homologous cloning method, and found that the contents of eicosenoic acid (20:1) and oleic acid (18:1) in the seeds of LPFATB transformed strains with Col-0 as the backgrounds were decreased to a certain extent, while the contents of palmitic acid (16:0) were significantly increased. In this study, the FATB enzyme gene was highly expressed in the flesh of Seedless and Tenera oil palm species and was positively correlated with the contents of free stearic acid and palmitic acid, while negatively correlated with the contents of free myristate acid. Dussert et al. [35] found that FATB in oil palm pulp can promote the synthesis of fatty acids, and EgWRI1-1 and EgWRI1-2 can combine FATB1, FATB1, and FATB3 to further promote the increase in fatty acid content in the TAG pathway. This result once again confirmed that the FATB gene is the key gene for fatty acid synthesis in oil palm pulp.

## 4. Materials and Methods

### 4.1. Plant Materials

In this experiment, the fruits of Seedless and Tenera at different developmental stages were selected from the experimental base of oil palm (19°33′ N, 110°47′ E) from the Institute of Coconut Research, Chinese Academy of Tropical Agricultural Sciences, Wenchang City, China. The pulp of each group at 95 days (early development stage: MS1 and MT1), 125 days (middle development stage: MS2 and MT2), and 185 days (late development stage: MS3 and MT3) after pollination were selected and frozen with liquid nitrogen and stored at −80 °C for the detection of metabolome and transcriptome. For each biological replicate, three independent technical replicates were analyzed for each different developmental stages.

### 4.2. Methods

#### 4.2.1. Metabolite Extraction

To determine the metabolite, 20 mg of sample was thawed into a centrifuge tube, containing 1 mL of lipid extract (methyl tert-butyl ether: methanol = 3:1, *v*/*v*, including internal standard mixture), and shaken for 30 min. Then, 300 μL of ultrapure water was added, shaken for 1 min, and placed at 4 °C for 10 min. Then, the tube was centrifuged for 3 min (12,000 rpm, 4 °C) and 400 μL of supernatant was transferred into a new centrifuge tube and concentrated at 20° C for 2 h, or until completely dry. Then, 200 μL of lipid complex solution (acetonitrile: isopropanol = 1:1, *v*/*v*) was added to the solution, and the solution was vortexed for 3 min and centrifuged for 10 min (12,000 rpm, 4 °C). Subsequently, 120 μl of the supernatant was transferred into the glass-lined tube until used.

#### 4.2.2. Metabolomics Analysis and Data Processing

The instrument platform for LC-MS/MS analysis was composed of ExionLC ultra-performance liquid chromatography in tandem with SCIEX QTRAP 6500+ mass spectrometer, and the chromatographic column was Thermo AccucoreC30 column (2.6 μm, 2.1 mm × 100 mm i.d.). The software Analyst 1.6.3 was used to process the mass spectral data and combined with the information of the local lipid database, mass spectral qualitative analysis was performed on the metabolites of the samples. The mass spectral peaks detected for the same metabolite in different samples were corrected, and the integrated peak areas of all samples detected were calculated to obtain the absolute content of metabolites. The MRM model results are shown in Appendix A. Differential metabolites between groups were screened using a combination of orthogonal partial least square discriminant analysis (OPLS-DA) and univariate analysis of *p*-value or Fold Change. The screening criteria were variable importance in projection (VIP) ≥ 1, Fold Change ≥ 2, or Fold Change ≤ 0.5 to obtain differential metabolites. The variation trend of different differential metabolite content can be obtained by analyzing the clustering heat map of differential metabolite content under Z-Score normalization treatment.

#### 4.2.3. Total RNA Extraction and High-Throughput Sequencing

RNA from oil palm pulp was extracted using the Plant Total RNA Extraction Kit for Tiangen Biota, and sequencing was performed using the Illumina HiSeq platform. Using HISAT2 [36], Clean Reads were sequentially compared with the reference genome (Genome assembly EG5 http://www.ncbi.nlm.nih.gov/assembly/GCA-000442705, (accessed on 27 January 2024)) to obtain location information on the reference genome or gene, as well as specific sequence characteristics of the sequenced samples. Gene expression was calculated using the FPKM (fragments per kilobase of transcript per million fragments mapped) method [37]. The quality of RNA-seq is shown in Appendix A.

#### 4.2.4. Transcriptome Data Processing and Differentially Expressed Genes Analysis

DESeq2/edgeR [38] was used to analyze the differentially expressed genes between the groups at different developmental stages, including MS1 vs. MS2, MS1 vs. MS3, MS2 vs. MS3, MT1 vs. MT2, MT1 vs. MT3, and MT2 vs. MT3; the screening threshold of the FDR (False Discovery Rate) is less than 0.05 with log2-Fold Change∣≥1. After that, the selected differentially expressed genes were analyzed by Nr functional annotation, enrichment analysis, and other bioinformatics.

#### 4.2.5. Combined Metabolome and Transcriptome Analysis

Kyoto encyclopedia of genes and genomes (KEGG) enrichment analysis: This experiment compares metabolite analysis results and genetic analysis results for the same gene, comparing the differences between the groups of metabolites and mapped to the KEGG pathway chart (https://www.genome.jp/kegg), can be found in the two groups to learn common enrichment pathway. Correlation analysis: The Pearson correlation coefficient was used to calculate the quantitative correlation between the absolute content of free fatty acid metabolites and the expression of related genes. Correlation results with a correlation coefficient > 0.80 and *p*-value < 0.05 were selected.

#### 4.2.6. Real-Time Fluorescence Quantitative PCR Verification

Ten genes with potential roles during anthocyanoside synthesis in oil palm pericarp were selected for real-time fluorescence quantitative PCR reactions. qRT-PCR primers were designed using NCBI (Appendix A) and Actin was used as an internal reference gene [39], and the QuantStudioTM7 Real-Time PCR Instrument was used along with the Real-Time Fluorescence Quantitative PCR Instrument to determine the relative expression of selected genes in 96-well microtiter plates. The relative expression of the selected genes was calculated using the 2^−ΔΔCt^ method. The qRT-PCR reaction system used in this study was 20 μL, consisting of SYBR^®^ Select Master Mix (2X): 10 μL, cDNA template: 1 uL, positive primer: 1 μL, water (without RNAase): 7 μL, and negative primer: 1uL. The reaction conditions were as follows: phase I UDG activation at 50 °C for 2 min; phase II UP activation at 95 °C for 2 min; phase III denaturation at 95 °C for 15 s for a total of 40 cycles; and phase Ⅳ annealing/extension at 60 °C for 1 min.

## 5. Conclusions

Oil palm fruit has the highest oil content at 185 days after pollination, which is the optimum harvesting time. Total free fatty acid levels in oil palm fruit are divided into three distinct phases: slow accumulation, rapid accumulation, and stable accumulation. These phases indicate different stages of development. During the slow accumulation phase, the levels of free fatty acids in the fruit gradually increase at a relatively slow pace. This is followed by the rapid accumulation phase, where the levels of free fatty acids increase rapidly, indicating a critical stage of development. Finally, the fruit enters the stable accumulation phase, where the levels of free fatty acids stabilize, indicating the fruit’s maturity. The expression patterns of key enzyme genes have a significant correlation with the synthesis and accumulation of free fatty acids. FabB was positively correlated with the contents of stearic acid, myristic acid, and palmitic acid and negatively correlated with the contents of free palmitic acid. FabF, an enzyme involved in fatty acid biosynthesis, has been found to exhibit a positive correlation with the levels of four main free fatty acids: stearic acid, myristate acid, palmitoleic acid, and palmitic acid. ACSL and FATB were positively correlated with the contents of three main free fatty acids (stearic acid, palmitoleic acid, and palmitic acid), and negatively correlated with the contents of free myristate acid. These results indicate that the expression of FabB and FabF enzyme genes may promote the synthesis of free myristate in oil palm flesh. The expression of FabF, ACSL, and FATB enzyme genes may promote the synthesis of free palmitoleic acid in oil palm flesh, and the expression of these four key enzyme genes may promote the synthesis of free stearic acid and palmitoleic acid in oil palm flesh. The expression of the FabB gene may inhibit the synthesis of stearic acid in oil palm flesh and the expression of the ACSL and FATB gene may inhibit the synthesis of myristate acid in oil palm flesh. The effects of these genes on the synthesis pathway of free fatty acids in oil palm were thoroughly examined. This research shed light on how these genes contribute to the production of free fatty acids, which are crucial components of oil palm fruit. Additionally, the study also investigated the dynamic evolution of free fatty acid contents in different stages of oil palm fruit development. By analyzing the changes in free fatty acid levels, researchers were able to gain a deeper understanding of the metabolic processes occurring during different developmental stages of oil palm fruit. This knowledge can be valuable for optimizing oil palm cultivation and improving the quality of oil palm products.

## Figures and Tables

**Figure 1 ijms-25-01686-f001:**
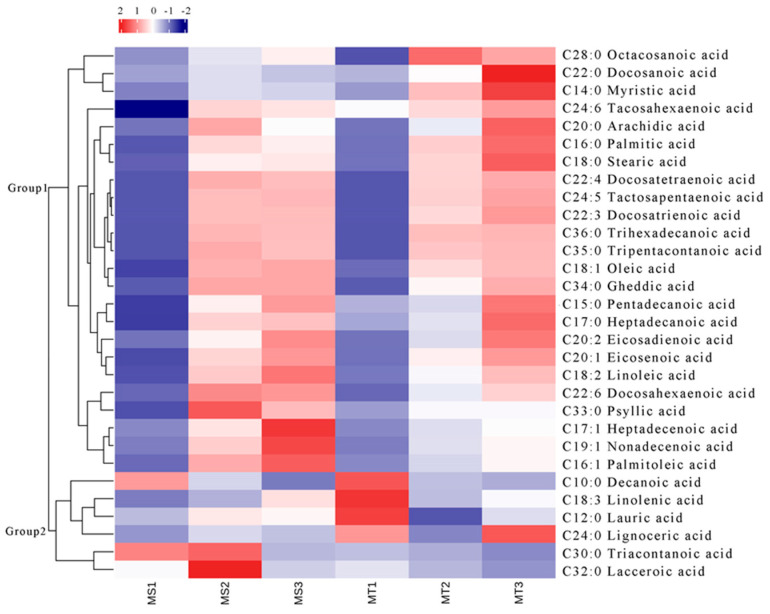
Heat map of metabolites clustering in the mesocarp of Seedless and Tenera oil palm during different developmental periods. The 6 columns in each heatmap represent the fruits development stages (MS1, MS2, MS3, MT1, MT2, and MT3).

**Figure 2 ijms-25-01686-f002:**
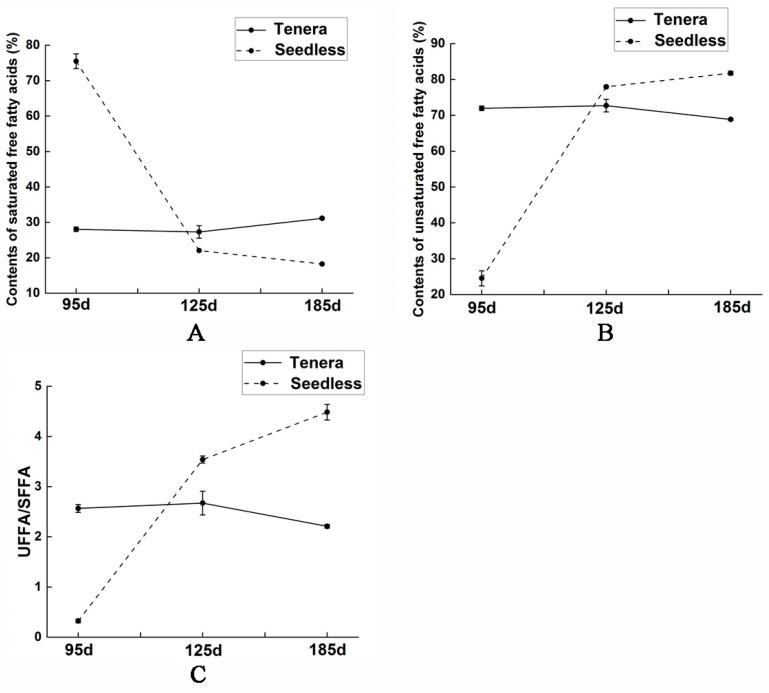
Dynamic changes of free fatty acid content in the mesocarp of Seedless and Tenera oil palm during different development periods. (**A**) changes in content of saturated fatty acids; (**B**) changes in content of unsaturated fatty acids; (**C**) changes in the content ratio of unsaturated fatty acids to saturated fatty acids. Seedless and Tenera oil palm during different developmental periods In summary, the study focused on analyzing the dynamic changes in fatty acid content in the flesh of two varieties of oil palm seeds. The result revealed substantial changes in fatty acid levels as the seeds developed from the early growth stage to maturity (MS1 to MS3 and MT1 to MT3). The proportion of total unsaturated free fatty acids in the flesh of Seedless and Tenera oil palm was found to be 81.74% and 68.87%, respectively, indicating that the total unsaturated free fatty acids in Seedless oil palm were found to be relatively high, with a higher degree of unsaturated palm oil compared to Tenera oil palms.

**Figure 3 ijms-25-01686-f003:**
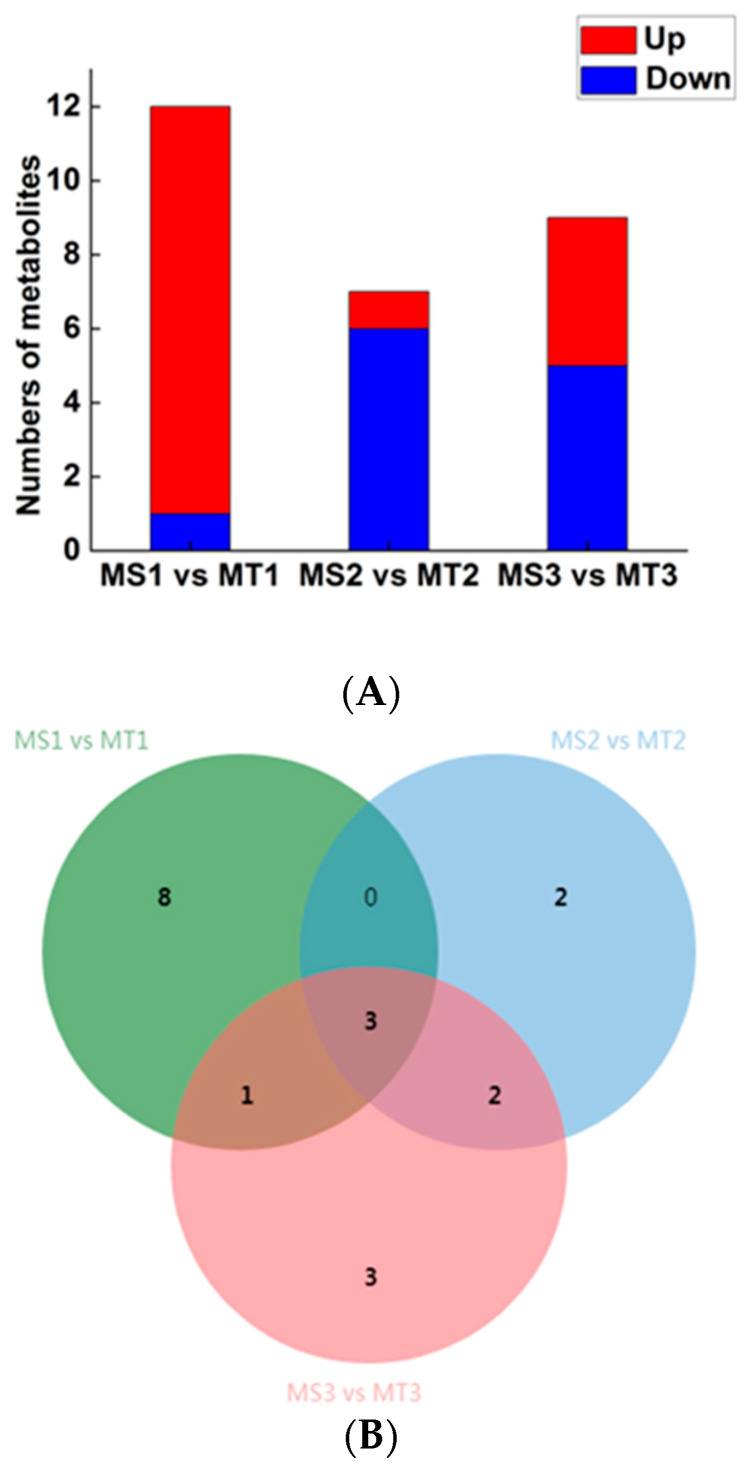
Oil palm differentially expressed metabolites statistics of MS and MT. (**A**) Venn diagram of differentially expressed metabolites; (**B**) histogram of significantly differentially expressed metabolites.

**Figure 4 ijms-25-01686-f004:**
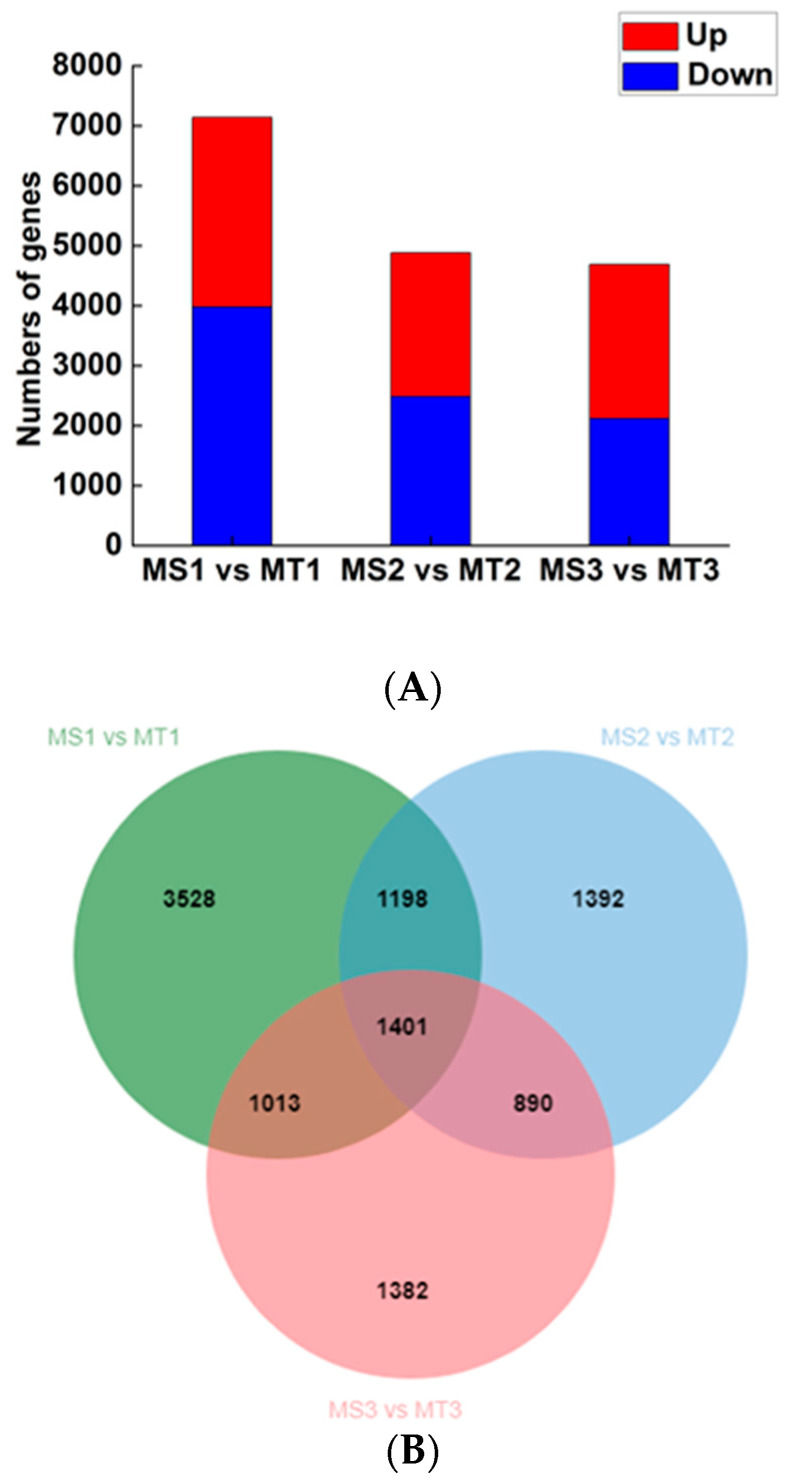
Oil palm differentially expressed genes statistics of MS and MT. (**A**) Venn diagram of differentially expressed genes; (**B**) histogram of significantly differentially expressed genes.

**Figure 5 ijms-25-01686-f005:**
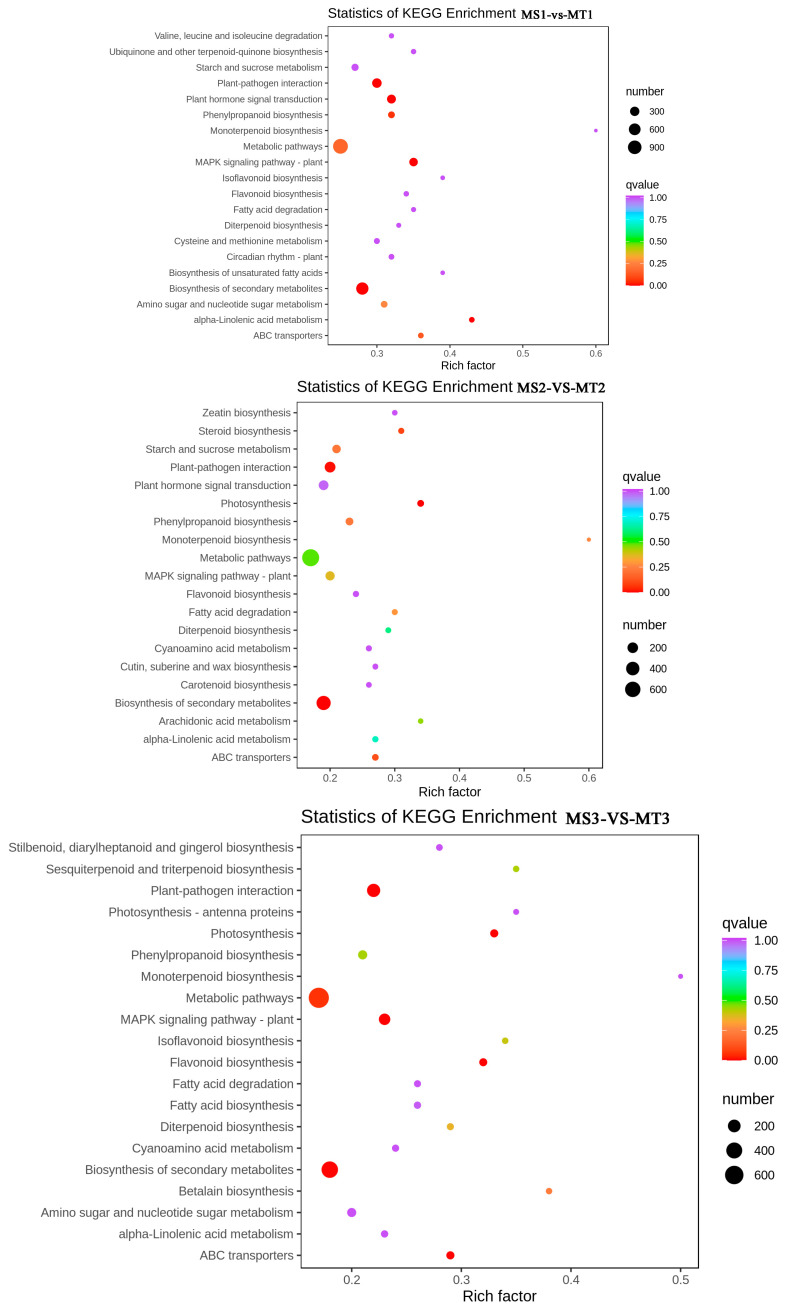
Top 20 of KEGG enrichment classification scatter chart for the pairwise comparisons of MS1 vs. MT1, MS2 vs. MT2, MS3 vs. MT3.

**Figure 6 ijms-25-01686-f006:**
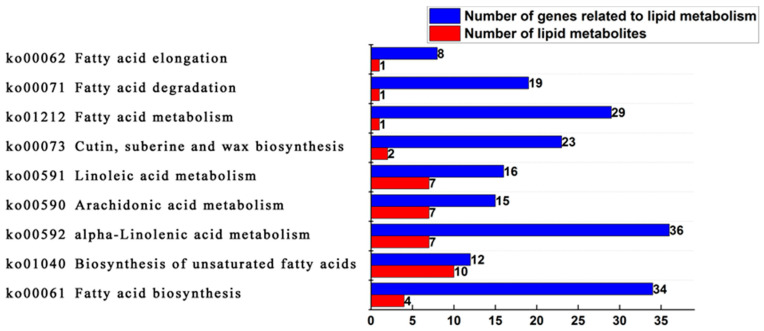
KEGG enrichment pathway statistics of lipid metabolites and genes related to lipid metabolism in the mesocarp of Seedless and Tenera oil palm during different developmental periods.

**Figure 7 ijms-25-01686-f007:**
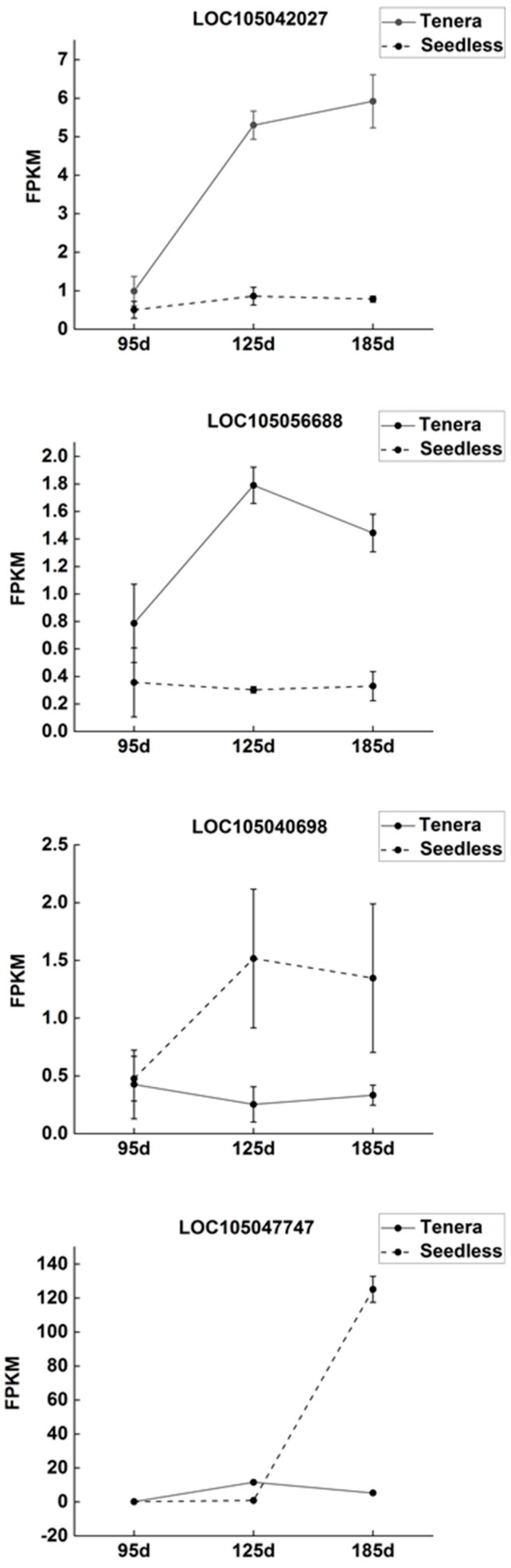
Dynamic changes in expression levels of key enzyme genes in the mesocarp of Seedless and Tenera oil palm during different developmental periods.

**Figure 8 ijms-25-01686-f008:**
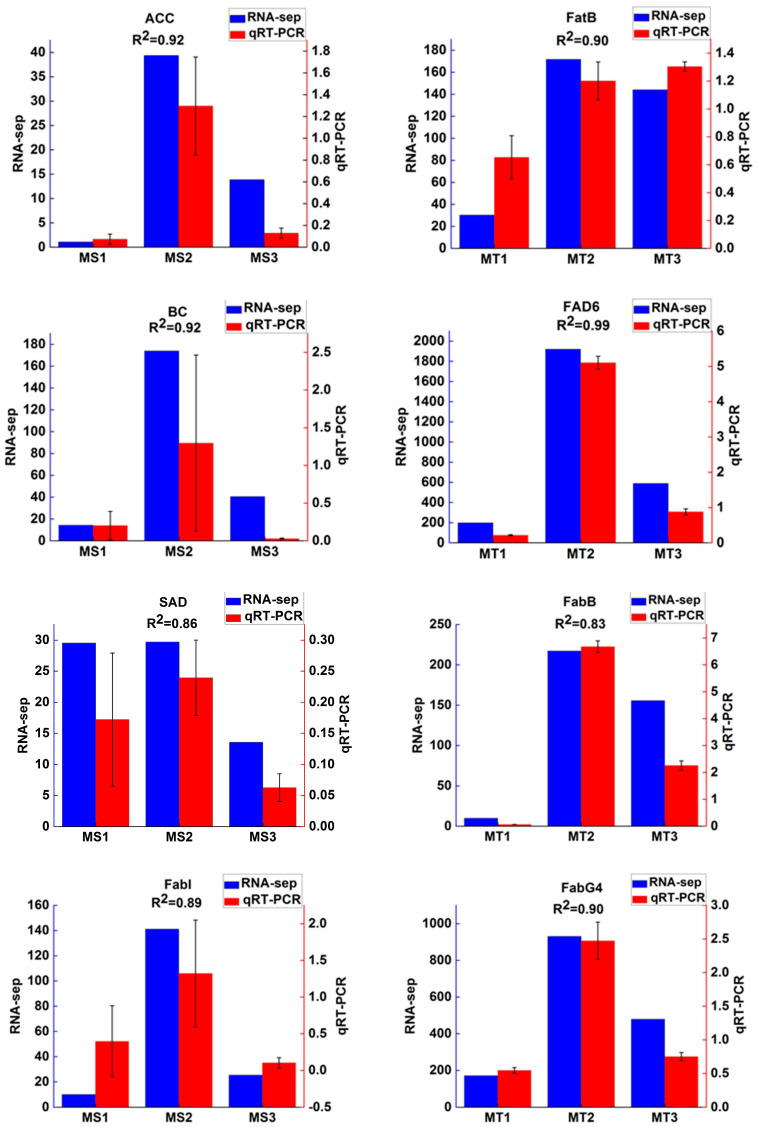
Relative expression levels of 10 selected genes during different developmental periods (MS1, MS2, MS3, MT1, MT2, and MT3). The 2^−ΔΔCt^ method was used to determine the relative expression levels of genes. The statistical differences were analyzed by ANOVA based on Duncan’s multiple test (*p* < 0.05).

**Figure 9 ijms-25-01686-f009:**
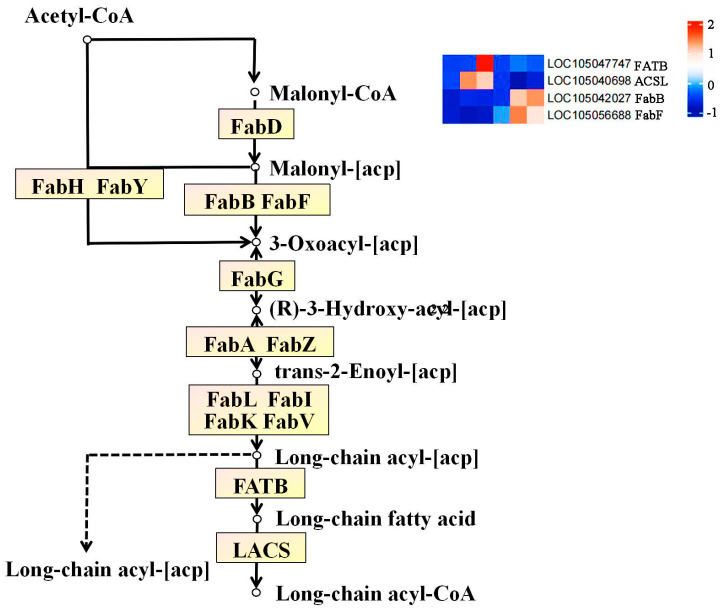
The main pathways and key enzymes of fatty acid biosynthesis. The yellow rectangle represents the enzyme. The rounded rectangle represents the connected channels. The white circle represents the compound.

**Table 1 ijms-25-01686-t001:** Dynamic changes in the accumulation of respective free lipid metabolites in the mesocarp of Seedless and Tenera oil palm during different developmental periods (nmol/g).

Free Lipid Metabolites	Different Developmental Periods					
		Seedless			Tenera	
	MS1	MS2	MS3	MT1	MT2	MT3
C10:0 Decanoic acid	375.30	266.61	206.91	428.91	253.11	239.25
C12:0 Lauric acid	6.52	9.45	8.91	15.32	3.47	7.54
C14:0 Myristic acid	20.10	112.27	101.73	44.33	225.33	371.80
C15:0 Pentadecanoic acid	6.49	26.89	36.51	17.39	21.18	40.47
C16:0 Palmitic acid	401.84	9126.63	8060.21	1401.43	9585.10	14,345.15
C17:0 Heptadecanoic acid	14.20	96.47	103.22	48.70	68.62	137.42
C18:0 Stearic acid	320.25	1684.93	1756.83	451.25	1922.86	2988.75
C20:0 Arachidic acid	\	133.86	75.33	\	65.60	178.70
C22:0 Docosanoic acid	35.44	47.38	42.29	39.38	54.47	104.06
C24:0 Lignoceric acid	53.07	69.62	63.93	110.51	49.04	128.63
C28:0 Octacosanoic acid	42.90	65.53	78.27	26.15	121.22	102.38
C30:0Triacontanoic acid	51.42	57.87	11.78	13.10	9.59	3.52
C32:0 Lacceroic acid	12.25	37.60	7.80	9.84	5.54	2.13
C33:0 Psyllic acid	2.25	12.34	9.23	4.32	6.86	6.87
C34:0 Gheddic acid	\	7.89	7.88	\	5.13	7.68
C35:0 Tripentacontanoic acid	\	37.46	33.83	\	32.80	34.80
C36:0 Trihexadecanoic acid	\	30.67	29.48	\	29.70	30.63
Total of saturated free fatty acid	1342.05	11,823.46	10,634.14	2610.62	12,460.62	18,729.78
C16:1 Palmitoleic acid	3.22	182.91	247.14	25.22	80.37	118.02
C17:1 Heptadecenoic acid	\	54.57	125.01	\	30.06	41.74
C18:1 Oleic acid	277.30	31,536.14	32,992.80	4549.25	26,021.06	30,396.86
C18:2 Linoleic acid	88.94	9508.93	13,475.84	1586.44	6680.46	10,188.00
C18:3 Linolenic acid	50.36	107.42	237.34	452.89	121.90	189.08
C19:1 Nonadecenoic acid	\	16.40	29.67	\	8.45	12.10
C20:1 Eicosenoic acid	\	301.53	395.73	47.56	260.78	392.81
C20:2 Eicosadienoic acid	\	2.07	3.67	\	1.38	3.96
C22:3 Docosatrienoic acid	\	19.36	19.44	\	16.65	22.56
C22:4 Docosatetraenoic acid	\	6.05	5.60	\	5.02	6.14
C22:6 Docosahexaenoic acid	\	10.09	9.55	\	4.56	7.15
C24:5 Tactosapentaenoic acid	\	9.49	9.76	\	8.60	10.66
C24:6 Tacosahexaenoic acid	11.23	47.89	45.71	41.35	47.22	55.90
Total of unsaturated free fatty acid	431.06	41,802.85	47,597.27	6702.71	33,286.52	41,444.97
Total of free fatty acid	1773.11	53,626.311	58,231.41	9313.33	45,746.14	60,174.75

Note: “\” indicate there are no or very little free fatty acid content in this period here; MS1 and MT1: early fruit development; MS2 and MT2: middle fruit development; MS3 and MT3: late stage of fruit development. Cluster analysis of all the free fatty acids in the flesh of Seedless and Tenera oil palm seeds at three different development stages. It showed that the overall clustering of 30 free fatty acids treated by Z-score normalization fell into 2 categories (Figure 1). In Group 1, there were 24 different kinds of free fatty acids. In the beginning, they were minimal and then surged rapidly, reaching their peak at the end of the development phase. The contents of heptadecenoic acid, nonadecenoic acid and palmitoleic acid in Seedless oil palm changed most obviously in MS1-MS3, but the contents of docosahexaenoic acid, tridecanoic acid, and arachidonic acid in MS1-MS3 increased first and then decreased in MS1-MS3, with the highest value in the middle stage of development (MS2). The contents of docosanic acid, myristic acid, and arachidic acid in Tenera oil palm changed particularly in MT1-MT3, but the contents of octacosanoic acid in MT1-MT3 increased first and then decreased, peaking in the middle stage of development (MT2), and the contents of tridecanoic acid in MT2-MT3 showed a stable trend. Its content reaches its highest value in MT3. In Group 2, the six free fatty acids changed in diverse ways. The content of decanoic acid in Seedless oil palm decreased continuously, while the quantity of linolenic acid in MS1-MS3 increased. In MS1-MS3, the levels of triaconic acid, lauric acid, tetracosanoic acid, and triededecanoic acid exhibited a decreasing trend followed by an increase, reaching their highest levels in MT3.

**Table 2 ijms-25-01686-t002:** Lipid metabolites and genes related to lipid metabolism statistics of the fatty acid biosynthesis (ko00061) pathway in the mesocarp of Seedless and Tenera oil palm during different developmental periods.

Comparable Group	Lipid Metabolites ID	Lipid Metabolites	Genes ID
MS1 vs. MT1	Lipid-B-N-0026	(FFA16:1) Palmitoleic acid	LOC105034507;LOC105034511;LOC105035520;LOC105039456;LOC105042279;LOC105061231;LOC105035641;LOC105040698;LOC105040700;LOC105044978;novel.3225;LOC105049274; ElguCp049;LOC105058068;LOC105037839; LOC105042027;LOC105056688;LOC105047747;LOC109506086;PTE;LOC105050310; LOC105048939
	Lipid-B-N-0005	(FFA14:0) Myristic acid	
MS2 vs. MT2	Lipid-B-N-0007	(FFA16:1) Palmitic acid	LOC105034507;LOC105034511;LOC105035520;LOC105039456;LOC105042279;LOC105033685;LOC105061231;LOC105056570;LOC105035641;LOC105040698;novel.3225;LOC105050555; ElguCp049;LOC105042027;LOC105056688; LOC105040257;LOC105047747;LOC109506086;PTE;LOC105048939
	Lipid-B-N-0009	(FFA18:0) Stearic acid	
	Lipid-B-N-0005	(FFA14:0) Myristic acid	
MS3 vs. MT3	Lipid-B-N-0026	(FFA16:1) Palmitoleic acid	LOC105034511;LOC105035520;LOC105058779;LOC105042279;LOC105035641;LOC105036494;LOC105039221;LOC105040698;LOC105032905;novel.3225;LOC105050555;ElguCp049; LOC105058068;LOC105051936;LOC105040922;LOC105042027;LOC105056688;LOC105040257;LOC105047747;LOC105050310;LOC105048939;LOC105049664;LOC105041644
	Lipid-B-N-0005	(FFA14:0) Myristic acid	

**Table 3 ijms-25-01686-t003:** Correlation analysis between the expression levels of key enzyme genes regulating free fatty acid synthesis and the contents of main free fatty acids in the mesocarp of Seedless and Tenera oil palm.

Gene ID	Enzyme Name	Stearic Acid	Myristic Acid	Palmitoleic ACID	Palmitic Acid
LOC105042027	FabB	0.75 **	0.917 **	−0.059	0.717 **
LOC105056688	FabF	0.517 *	0.722 **	0.252	0.490 *
LOC105040698	ACSL	0.060	−0.256	0.697 **	0.123
LOC105047747	FATB	0.167	−0.106	0.723 **	0.134

Note: “*” indicates significant correlation (*p* < 0.05), and “**”indicates extremely significant correlation (*p* < 0.01).

## Data Availability

Data are contained within the article.

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
