# Peer review of "Metabonomics and Transcriptomic Analysis of Free Fatty Acid Synthesis in Seedless and Tenera Oil Palm"

_ijms, 2024, doi:10.3390/ijms25031686_

Round 1
Reviewer 1 Report
Comments and Suggestions for Authors
The authors should specify exactly the material they worked on. A reader who is not familiar with palm taxonomy does not know what the terms mean: "Seedless and Tenera". You can also cite in this part: Oil Palm and Coconut S. A. C. N. Perera https://link.springer.com/chapter/10.1007/978-1-4614-9572-7_11
Why did the authors use Tenera variety when this variety was already analyzed, please see https://www.frontiersin.org/articles/10.3389/fpls.2023.1132024/full
Why don't the authors write that a very similar paper on this topic has been published see Shuyan Zhang et al. Differential analysis of transcriptomic and metabolomic of free fatty acid rancidity process in oil palm (Elaeis guineensis) fruits of different husk types.
Why don't the authors cite the work: Dussert et al. Comparative Transcriptome Analysis of Three Oil Palm Fruit and Seed Tissues That Differ in Oil Content and Fatty Acid Composition https://academic.oup.com/plphys/article/162/3/1337/6110847
I expect the authors to study the literature carefully and show what data they obtained is new! Please refer in detail to the works I mentioned.
As for the references, the authors made a lot of minor mistakes when it comes to writing the names of genera and species (all should be in italics).
Reviewer 2 Report
Comments and Suggestions for Authors
In this study, the authors analyzes the changes in free fatty acid content and key enzyme gene expression in Seedless and Tenera oil palm fruits after pollination. It speculates on the regulatory effects of key enzyme genes on free fatty acid, providing valuable data for research on fatty acid synthesis mechanisms, improving palm oil quality, and accelerating industry development. In this experiment, the fruits of Seedless and Tenera at different developmental stages were selected from the experimental base of oil palm of the Institute of Coconut Research, Chinese Academy of Tropical Agricultural Sciences, The pulp of each group at 95 days (early stage develop ent), 125 days (middle development stage) and 185 days (Late development stage) after pollination were selected and frozen with liquid nitrogen and stored at -80℃ for the detection of metabolome and transcriptome. For each biological replicate, three independent technical replicates were analyzed for each different developmental stages. The results showed that the total free fatty acid content of oil palm changed significantly before and after the development of the flesh, and the total free fatty acid content gradually increased with the continuous growth and development of the flesh, and the total free fatty acid content increased sharply in the early to middle stages of development. The total free fatty acid content of seedless oil palm is stable from mid to late development. However, the most significant changes were observed in the contents of linolenic acid, linoleic acid, palmitic acid and eicosenoic acid in different development stages.
The paper is interesting but I have some remarks:
- The figures are not clear (e.g. fig. 5)
- The authors have to improve the references section
- Extensive editing of English language required
Comments on the Quality of English LanguageExtensive editing of English language required
Round 2
Reviewer 1 Report
Comments and Suggestions for Authors
The authors took into account my comments and suggestions and modified the article. Thank you.
I suggest reviewing the article again for language and correctness of references.
I would suggest adding to the introduction with two-three sentences that oil palm cultivation is somewhat controversial for ecological and ethical reasons. E.g.: Wilcove, David S.; Koh, Lian Pin. Addressing the threats to biodiversity from oil-palm agriculture. Biodiversity and Conservation. 19 (4): 999–1007. doi:10.1007/s10531-009-9760 Fitzherbert, Emily B.; Struebig, Matthew J.; Morel, Alexandra; Danielsen, Finn; Brühl, Carsten A.; Donald, Paul F.; Phalan, Ben. How will oil palm expansion affect biodiversity? Trends in Ecology & Evolution. 23 (10): 538–545. doi:10.1016/j.tree.2008.06.012 Koh, Lian Pin; Wilcove, David S. (2008). Is oil palm agriculture really destroying tropical biodiversity?. Conservation Letters. 1 (2): 60–64. doi:10.1111/j.1755-263X.2008.00011.xAuthor Response
Comments and Reply
The authors took into account my comments and suggestions and modified the article. Thank you.
Comment: I suggest reviewing the article again for language and correctness of references.
Reply: Thanks for your suggestion. We have various factors such as grammar and sentence structure. We ensure that the language is clear, concise, and properly conveys the intended message.We have corrected the mistakes in reference section.
Comment: I would suggest adding to the introduction with two-three sentences that oil palm cultivation is somewhat controversial for ecological and ethical reasons. E.g.: Wilcove, David S.; Koh, Lian Pin. Addressing the threats to biodiversity from oil-palm agriculture. Biodiversity and Conservation. 19 (4): 999–1007. doi:10.1007/s10531-009-9760 Fitzherbert, Emily B.; Struebig, Matthew J.; Morel, Alexandra; Danielsen, Finn; Brühl, Carsten A.; Donald, Paul F.; Phalan, Ben. How will oil palm expansion affect biodiversity? Trends in Ecology & Evolution. 23 (10): 538–545. doi:10.1016/j.tree.2008.06.012 Koh, Lian Pin; Wilcove, David S. (2008). Is oil palm agriculture really destroying tropical biodiversity?. Conservation Letters. 1 (2): 60–64. doi:10.1111/j.1755-263X.2008.00011.x
Reply: Thanks for your suggestion. Detailed explanation on ecology and biodiversity of oil palm were included (Line No: 46 - 55).